# Timing the escape of a photoexcited electron from a molecular cage

Connor Fields[1], Aleksandra Foerster[2], Sadegh Ghaderzadeh [2], Ilya Popov[2], Bang Huynh [2], Filipe Junqueira[1], Tyler James [1], Sofia Alonso Perez[1], David A. Duncan [2,3], Tien-Lin Lee [3], Yitao Wang[1], Sally Bloodworth [4], Gabriela Hoffman[4], Mark Walkey[4], Richard J. Whitby[4], Malcolm H. Levitt [4], Brian Kiraly [1], James N. O'Shea[1], Elena Besley [2] ✉ & Philip Moriarty [1] ✉

Charge transfer is fundamentally dependent on the overlap of the orbitals comprising the transport pathway. This has key implications for molecular, nanoscale, and quantum technologies, for which delocalization (and decoherence) rates are essential figures of merit. Here, we apply the core hole clock technique—an energy-domain variant of ultrafast spectroscopy—to probe the delocalization of a photoexcited electron inside a closed molecular cage, namely the Ar $2p^54s^1$ state of Ar@$C_{60}$. Despite marginal frontier orbital mixing in the ground configuration, almost 80% of the excited state density is found outside the buckyball due to the formation of a markedly diffuse hybrid orbital. Far from isolating the intracage excitation, the surrounding fullerene is instead a remarkably efficient conduit for electron transfer: we measure characteristic delocalization times of $6.6 \pm 0.3$ fs and $\lesssim 500$ attoseconds, respectively, for a 3D Ar@$C_{60}$ film and a 2D monolayer on Ag(111).

In an influential and enduring paper[1], Roald Hoffmann laid out a set of core principles associated with the interaction of localized orbitals in molecular systems, with a particular focus on the balance of through-space and through-bond coupling. Over fifty years later, Hoffmann's insights not only continue to underpin a great deal of what is now essentially seen as chemical intuition, but multidisciplinary fields of research such as molecular electronics, photovoltaic/solar cell development (and photochemistry/photophysics in general), surface science, and nanoscience all owe a great deal to his work.

Alongside what might be best described as the static coupling of orbitals explored by Hoffmann, a central focus of each of those fields— molecular electronics in particular—is the measurement, control, and exploitation of the tunnelling of carriers between, and through, units, contacts, and spacers in molecular and nanoscale architectures[2]. In other words, it is the dynamic properties of charge delocalization and motion[3–5], via mechanisms such as resonant, non-resonant, or super-exchange tunnelling, thermally-dependent diffusive transport, and/or

variable range hopping that are of especial interest[6–8]. These in turn determine the electrical conductance of a molecular or nanoscale component/junction, as described, for example, by the Landauer-Buttiker formalism (and subsequent modifications thereof)[9,10].

We focus here on a molecular system that is unique in the context of through-space versus through-bond transport: endohedral fullerenes. Although their host-guest nature is of course not without chemical parallel[11–13], no other chemical system—including clathrates, inclusion complexes, zeolites, metal-organic frameworks, and supramolecular assemblies—involves total encapsulation and containment inside a "seamless" framework, where the guest species cannot leave without covalent bonds being broken, as is the case for endofullerenes. This has critical implications in terms of the isolation of the encapsulate from its surrounding physicochemical environment and, as we shall see, for the dynamics of charge transfer to/from the encaged species.

In this context, and despite what might be described as its chemical oddness, Ar@$C_{60}$—a single argon atom encapsulated within a

[1]School of Physics & Astronomy, University of Nottingham, Nottingham, UK. [2]School of Chemistry, University of Nottingham, Nottingham, UK. [3]Diamond Light Source, Harwell Science & Innovation Campus, Didcot, UK. [4]School of Chemistry and Chemical Engineering, University of Southampton, Southampton, UK. ✉e-mail: elena.besley@nottingham.ac.uk; philip.moriarty@nottingham.ac.uk

$C_{60}$ cage, Fig. 1a[14,15]—is a particularly intriguing limiting case. In the ground state, there is remarkably little hybridization of the encapsulated Ar with the frontier $C_{60}$ orbitals (i.e., highest occupied molecular orbital (HOMO), lowest unoccupied molecular orbital (LUMO), HOMO-1, LUMO+1 etc.) Although Morscher et al.[16] provide compelling evidence for a hybrid Ar $3p$-$6T_{1u}$ state, this is located 8 eV below the HOMO binding energy, i.e., ~10 eV below the Fermi level, and therefore well outside the energy range for electron transfer that underpins conductance in molecular electronics architectures. Given the marginal ground state coupling of the Ar atom with the fullerene frontier orbitals, one might ask whether this lack of overlap extends to excited states inside the cage. We have therefore measured the delocalization rate of a photoexcited state of the encapsulated argon.

In this work, we use the Auger-Meitner resonant Raman variant[17] of the core hole clock technique[18,19] to monitor, with sub-femtosecond temporal resolution, the delocalization of a photoexcited Ar $4s$ electron (Ar $2p_{3/2} \rightarrow 4s$) for Ar@$C_{60}$ molecules adsorbed as a bulk film or as a monolayer on a Ag(111) surface. For the latter, we complement the resonant Auger-Meitner analysis with normal incidence X-ray standing wave (NIXSW)[20,21] measurements, enabling, in parallel, an accurate determination of the position of the Ar atom above the substrate. We find that the naïve picture of decoupled Ar and fullerene orbitals outlined above entirely fails to explain the electron delocalization dynamics that occur in the endofullerene system. Density functional theory (DFT) calculations combined with the maximum overlap method (MOM)[22,23] reveal that the photoexcited state is exceptionally diffuse, with ~80% of its density delocalized outside the cage. The hydrogenic superatom orbital (SAMO) states of fullerenes, first proposed by Feng et al.[24,25], are a compelling candidate for the origin of the extensive delocalization.

## Results and discussion

The core-hole clock (CHC) technique[19,26–28], first introduced in the early nineties[18,29], is an energy-domain alternative to ultrafast pump-probe spectroscopy that is capable of measuring the rate of electron transfer on time scales ranging from tens of attoseconds[30] to ~100 femtoseconds (depending on the lifetime, $\tau_{CH}$, of the particular core hole that is used as the clock[31]). CHC spectroscopy also has the key advantage of being chemically specific, with all of the attendant spectral finger-printing advantages; this aspect is pivotal for the work described herein.

A schematic of the CHC protocol used to determine the rate of delocalization of the photoexcited Ar $4s$ state in Ar@$C_{60}$ is shown in Fig. 1b. Resonant X-ray excitation from the Ar $2p_{3/2}$ level produces an initial core-excited Ar $2p^5 4s^1$ configuration. There are then two primary channels for the subsequent decay of that excited state: (i) a spectator

Auger-Meitner process, where the $4s$ electron does not delocalize before decay of the core excitation, and (ii) traditional Auger-Meitner electron emission, where the photoexcited electron has tunnelled away from the original excitation site before core-hole decay. (Note that, as demonstrated by Fig. 2c, the participator (or resonant photoemission) decay channel plays a negligible role in the case of Ar@$C_{60}$.) A key assumption here is that the core hole decay and electron delocalization rates are independent of each other. Moreover, both are assumed to follow a first order rate equation[19,26] and therefore decay with an exponential dependence.

We focus on X-ray absorption across the Ar $2p_{3/2} \rightarrow 4s$ spectral peak. This is not only the initial excitation step in the CHC process (Fig. 1b), but the X-ray absorption spectrum by itself (Fig. 2a) already provides a great deal of insight into the degree of electronic coupling of the encapsulated Ar with the surrounding environment. The Ar $2p_{3/2} \rightarrow 4s$ absorption spectrum shown in Fig. 2a (for a bulk film of Ar@$C_{60}$) is best fitted with a pure Lorentzian function, whose linewidth (full width at half-maximum (FWHM)) of 280 (±10) meV should be compared with the ~120 meV linewidth of gas phase argon[32,33]. Figure 2a is a line-by-line integral of the resonant Auger-Meitner map of Fig. 2b, i.e., it is a partial electron yield X-ray absorption measurement. (A fit to a total electron yield XAS spectrum (Supp. Fig. 2) results in a linewidth that agrees within experimental uncertainty, 260 ± 10 meV, with that of the partial yield spectrum).

In the solid state, the extensive X-ray absorption, CHC, and photoemission measurements of argon on variously adsorbed graphene (Gr) monolayers reported by Lizzit et al.[34] arguably represent the most appropriate dataset with which to compare our Ar@$C_{60}$ XAS and CHC results. As described below, the Ar@$C_{60}$ system surprisingly exhibits behaviour at odds with that for the weakly coupled Gr/O/Ru, Gr/SiO₂, and Gr/SiC systems (i.e., unlike that expected for an isolated argon atom).[34]

### Quantifying the electron delocalization time

By decomposing the decay spectrum into its normal Auger-Meitner and spectator components (Fig. 2), the characteristic delocalization time (often simply called the charge transfer time), $\tau_D$, for the X-ray excited Ar $4s$ electron can be determined[26] (see "Methods", Supp. Note 3, Supp. Fig. 3, and Supp. Fig. 4). When adsorbed directly on a metal, the value of $\tau_D$ measured in this way for argon is of order a few fs[35,36]; with a graphene monolayer sandwiched between the metal and argon, the on-resonance value of $\tau_D$ varies from ~3 fs to 16 fs (depending on the level of graphene-metal interaction)[34]; and for argon decoupled from the substrate via an underlying Ar/Xe spacer layer, the value of $\tau_{CT}$ increases to over 50 fs[35]. (Indeed, the calculations of Gauyacq and Borisov[37] predict values of $\tau_D$ as large as 7 picoseconds

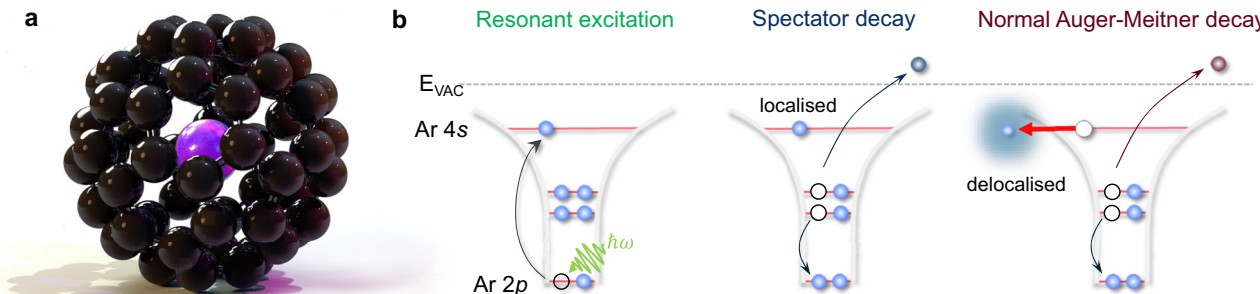

**Fig. 1 | Core-hole clock spectroscopy of an endofullerene. a** 3D-rendered image of the ground state geometry of Ar@$C_{60}$ predicted by density functional theory (see "Methods"). **b** The core-hole clock technique. Following resonant excitation via X-ray absorption, the photo-excited Ar $2p^5 4s^1$ state can decay either via a spectator Auger-Meitner process, where the $4s$ electron remains localized on the time scale of the core hole decay, or a normal Auger-Meitner process, for which the $4s$ electron has tunnelled away (into the surrounding molecular matrix and/or substrate) before the core hole decays. The relative intensity of electron emission via these channels enables the delocalization rate of the $4s$ state to be determined.

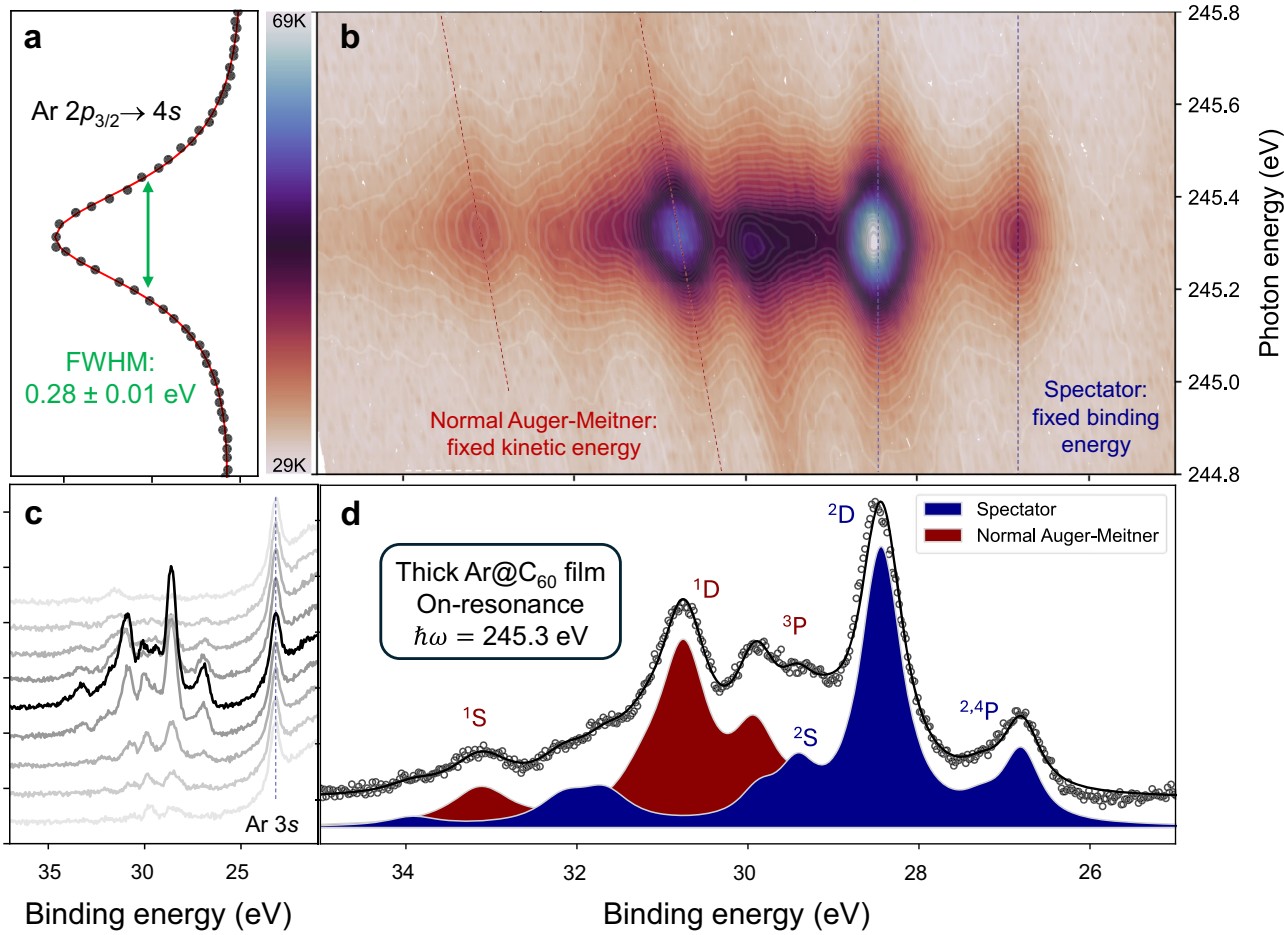

**Fig. 2 | Timing electron delocalization in Ar@C₆₀. a** Ar $2p_{3/2} \to 4s$ partial electron yield X-ray absorption spectrum (filled circles) with Lorentzian fit (red line). The photon energy axis runs vertically and is the same as that for the resonant Auger-Meitner map shown in (**b**). The intensity axis (arbitrary units) runs horizontally. **b** Resonant Auger-Meitner map showing intensity of decay spectra as a function of photon energy. (Intensity is shown as a false colour map. While the units of this false colour scale are arbitrary, the values recorded by the CCD detector are directly proportional to the Auger-Meitner electron count rate. The CCD values span a range of ~29,000 to ~69,000 arbitrary units, i.e. the "29K" and "69K" limits shown on the colour bar alongside the map). The X-ray absorption spectrum shown in (**a**) both shares its photon energy axis with the resonant Auger-Meitner map and is also the line-by-line integral of the map. (A total electron yield spectrum is included in the Supplementary Information (Supp. Fig. 2)). Normal Auger-Meitner transitions are at fixed kinetic energy and therefore disperse diagonally with photon energy

(red dashed lines), whereas the spectator peaks are at fixed binding energy (blue dashed lines). For clarity, the dispersion (or lack thereof) of all peaks is not shown. **c** Auger-Meitner electron spectra spanning ± 400 meV either side of the resonance condition in 100 meV increments. The on-resonance spectrum is plotted in black; the dashed blue line highlights the Ar 3s peak position. The Ar 3s peak intensity does not resonate and is constant to within 5%, highlighting the negligible contribution of participator transitions. (Note that the map in (**b**) and the set of spectra in (**c**) were acquired from different, but similarly prepared, Ar@C₆₀ samples.) **d** On-resonance decay spectrum showing decomposition into the various normal Auger-Meitner and spectator components. Following Karis et al.[64], we associate the shake-up features at ~ 32 eV and ~ 34 eV binding energy with spectator intensity of $3p^4 5s^1$ character. See Supp. Note 3 for more details on the fitting process. The map in (**b**) shares its binding energy axis with the on-resonance spectrum. Source data are provided as Source Data files.

for thick argon films.) In the context of the Ar@C₆₀ system where there is marginal mixing of the argon and fullerene density in the ground state, one might initially, and perhaps naïvely, expect the charge transfer rate to be relatively slow—comparable, at least, to that for the decoupled and weakly interacting Ar-on-graphene and Ar-on-Xe systems. (Although see the section titled *Role of the Z + 1 approximation* below.) This is not at all what we find.

Despite the apparent chemical isolation of the encapsulated argon atom within the fullerene cage, the on-resonance value of $\tau_D$ for bulk Ar@C₆₀, 6.6 (±0.3) fs, shows that not only does electron delocalization occur on a time scale that is up to three orders of magnitude faster than that predicted for bare argon atoms condensed in a thick multilayer film (see, for example, Table II of Gauyacq and Borisov[37]) but that the charge transfer rate is comparable to that for argon separated from a metal substrate (namely Pt(111)) by a graphene monolayer[34], despite the Ar@C₆₀ solid having a band gap larger than 2 eV. Moreover, the primary trend of a reduction in delocalization time as a

function of increasing photon energy (Supp. Fig. 4) is entirely opposite to that observed for argon adsorbed directly on a variety of metal surfaces (including Ag(111)), where the band structure of the substrate (and the concomitant wave-vector matching requirement) leads to larger values of $\tau_D$ as $\hbar\omega$ is increased[38].

These observations all point to a substantial coupling and mixing of the core-excited Ar 4s state with the surrounding carbon cage, rather than an isolation of the excited state within the endofullerene. To interpret this mixing of the argon and fullerene density, and to gain a deeper understanding of the concomitant rapid transfer of the photoexcited 4s electron, we turn to quantum chemistry calculations.

### Beyond the confines of the cage: Ar 4s delocalization

Despite the seeming lack of any interaction beyond dispersion forces in the endofullerene crystal (a van der Waals solid), there is clearly a relatively facile delocalization pathway available to the photoexcited Ar 4s electron. Excited-state calculations exploiting the maximum

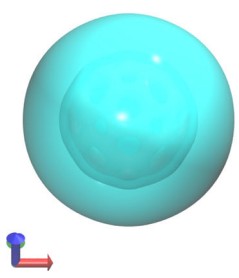

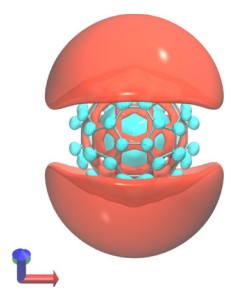

### ground-state Ar-based *4s*

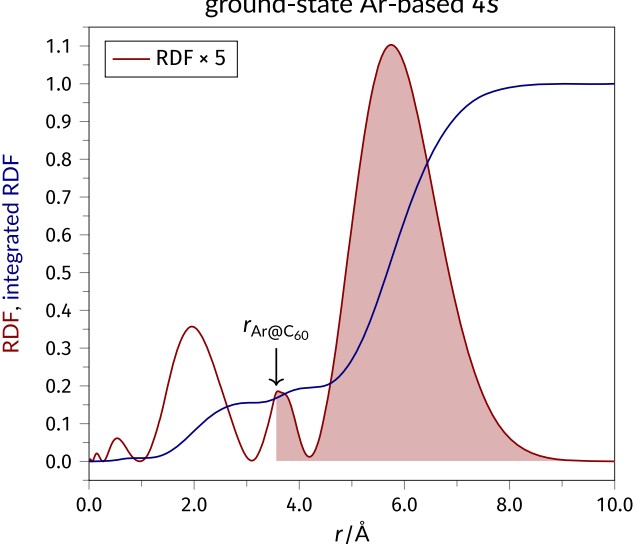

### excited-state distorted *4s*

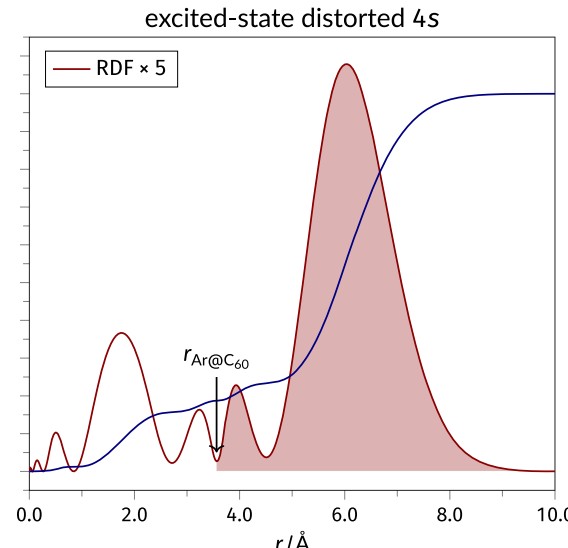

**Fig. 3 | Ground vs core-excited *4s* state.** Isosurfaces and radial distribution functions for the ground-state and excited-state *4s* molecular orbitals. The isosurfaces are plotted at isovalues of $\pm 0.04\,\text{Å}^{-3}$. The shaded area of each radial distribution function highlights the density that exists beyond the radius of the cage, $r_{\text{Ar@C}_{60}}$. Atomic cordinates are provided as Supplementary Data 2.

overlap method (MOM) [22,23] (see "Methods" and Supp. Note 8) provide key insights into the rapid escape of the encaged Ar *4s* electron. (A justification of our use of the MOM, and a comparison with time-dependent density functional theory calculations, is given in Supp. Note 8. We also discuss relativistic considerations in Supp. Note 9).

Figure 3 shows isosurfaces and radial distribution functions for the ground-state and excited-state *4s* orbitals, with the latter calculated using the MOM. To estimate the spatial extent of the ground and photoexcited states, we have integrated the spherically averaged radial density distribution and determined the fraction of the density that is found at distances larger than the cage radius. For both the ground and excited states, the *4s* orbital extends significantly beyond the confines of the fullerene cage, with more than 80% of the density lying beyond the Ar@C$_{60}$ radius of 3.54 Å. However, while the ground state unoccupied *4s* orbital is of almost exclusively argon character (92% contribution), the excited state instead has only a 13% Ar contribution. In other words, the highly delocalized excited state is of majority carbon, i.e., fullerene cage, character. (We use the C-squared population analysis method of Ros and Schuit[39] to determine the contributions. See S.I. for a detailed discussion).

The combination of the dominant fullerene character and the highly diffuse nature of the excited state is characteristic of superatomic molecular orbitals[24,25,40]. Conceptually similar to, but distinct from, Rydberg orbitals, superatomic orbitals[24,25,40] are not bound to the carbon atoms of the fullerene cage (unlike the traditional HOMO, LUMO etc.). Instead, SAMO states arise from the central potential of the core of the buckyball and are unique to hollow molecules; just as for the hydrogen atom, SAMO wavefunctions correspond to different orbital angular momentum states (*s*, *p*, *d*...) Of particular relevance to the interpretation of our core hole clock results, SAMO wavefunctions extend far beyond the carbon-atom-derived $\sigma$ and $\pi$ orbitals, to the extent that hybridization into metal-like nearly-free-electron bands occurs, with a substantial bandwidth (~ 600 meV) in the bulk fullerite crystal[40]. In the context of electron transfer, this represents a new and fascinating addition to Hoffmann's schema: interaction and delocalization via a coupling of atomic and superatomic orbitals.

With a SOMO-mediated delocalization process in mind, we have used the QSYM[41] framework to examine the symmetry of the excited-state orbital. Applying the relevant QSYM$^2$ projection operators (see Supp. Note 7), we find that the excited-state orbital shown in Fig. 3b comprises approximately 76% *S*-symmetry component, 23% *D*-symmetry component, and a very small contribution (less than 1%) from *G*-symmetry. This is to be contrasted with the ground state *4s* orbital, which has essentially pure (i.e., almost 100%) *S*-symmetry. The significant incorporation of the *D*-symmetry component in the excited *4s* orbital is attributed to the interaction with the carbon cage (as expected from the population analysis discussed above), providing a mechanism for mixing of argon and fullerene density in a highly delocalized orbital.

Consideration of the relative energies of the levels underpinning the core hole clock experiment broadly supports the proposal of delocalization via mixing with SAMO density. By comparison with the Ar $2p_{3/2}$ core level and HOMO binding energies, both referenced to the Fermi level ($E_F$), we find that the Ar $2p_{3/2} \rightarrow 4s$ resonance is located $4.80 \pm 0.15$ eV above the HOMO level (Supp. Notes 1 and 2 and Supp. Figs 1 and 2). This agrees remarkably well—although see Supp. Note 2

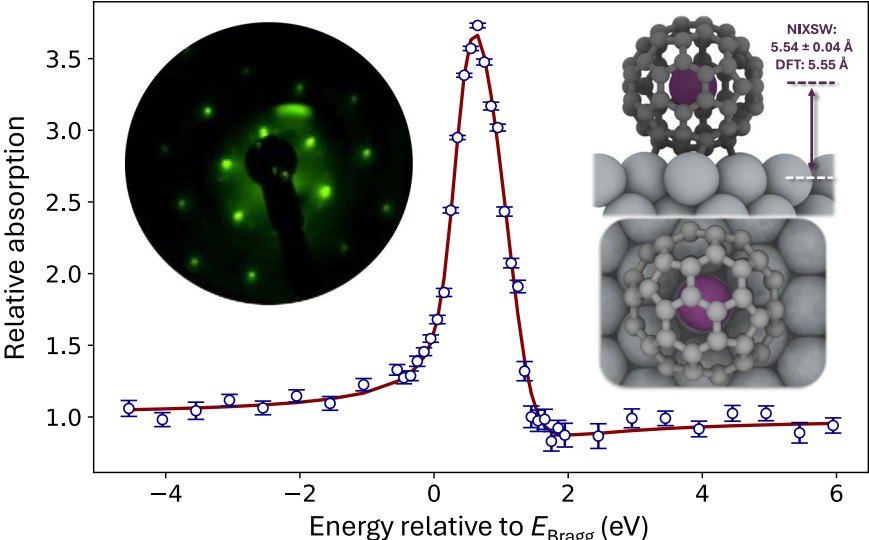

**Fig. 4 | Locating the argon atom in adsorbed Ar@C$_{60}$.** The normal incidence X-ray standing wave (NIXSW) profile derived from the variation in the Ar 2$p_{3/2}$ photoemission yield for an Ar@C$_{60}$ monolayer on Ag(111) is shown as the blue open circles in the main plot. A least squares fit to this profile (red line) (see "Methods") yields an Ar-Ag(111) separation of 5.54 ± 0.04 Å, placing the Ar atom at the centre of the cage, despite the strong interaction of the surrounding fullerene with the Ag(111) surface. This separation is identical within experimental uncertainty to the Ar-Ag(111) adsorption height of 5.55 Å predicted by our **(inset to right)** ground state PAW-DFT calculations for the 6:6 on-top geometry of the fullerene cage. The error bars for the data points comprising the NIXSW profile were calculated from the uncertainty in the fitted integrated intensity of the Ar 2$p_{3/2}$ core-level peak across the photon energy range. That uncertainty in turn is derived from the diagonalisation of the covariance matrix output when the Levenberg-Marquardt fitting routine converged. We used the `lmfit` Python package for fitting. **Inset to left:** $(2\sqrt{3} \times 2\sqrt{3})R30^{o}$ LEED pattern for the Ar@C$_{60}$ monolayer. Source data are provided as Source Data files and atomic coordinates for the DFT calculations are provided as Supplementary Data 1.

for a discussion of the role of core/valence excitons—with the band structure calculations of Zhao et al.[40], which place the centre of the s-SAMO band at 4.8 eV above the HOMO level (i.e., resonant with the (core-excited) Ar 4s energy).

We also note that our measured value of the Ar 2$p$ → 4$s$ X-ray absorption resonance for ~1 monolayer (ML) coverage (see below) of Ar@C$_{60}$ on Ag(111), viz. 3.2 ± 0.1 eV above $E_F$, is identical to that reported by Dutton et al.[42] for the position of the s-SAMO resonance of the 1 ML C$_{60}$/Ag(111) system. Moreover, our measured on-resonance value of $\tau_D = 6.6 \pm 0.3$ fs for the bulk Ar@C$_{60}$ film is entirely in line with the 4–20 fs range for the s-SAMO lifetime (for empty C$_{60}$) determined by Zhu et al.[43].

### Role of the Z+1 approximation

Argon in the core-excited $2p^5 4s^1$ state is chemically very similar to ground state potassium: this is the well-known Z + 1 approximation[44,45] used extensively to interpret core level spectra and core hole clock measurements. Given that, in turn, potassium readily dopes the LUMO of C$_{60}$[46], the rapid delocalization of the photoexcited Ar 4$s$ electron that we observe could possibly arise from transient doping of the LUMO (although we highlight that, as discussed in the preceding paragraph, the Ar 4$s$ resonance lies significantly above the LUMO energy). We have investigated this K-doping possibility at length using ground state DFT calculations.

We computed the ground state of K@C$_{60}$ at the PBE/6-31++G** level. Our results are very similar to those previously reported by Östling and Rosén[47]. In particular, we find electron transfer from the encapsulated K atom to the C$_{60}$ LUMO, resulting in the occupation of one of the previously vacant $t_{1u}$ molecular orbitals and the close-to-complete (98%) deoccupation of the K 4$s$ level (Supp. Fig. 9 and Supp. Note 10). However, in addition to the energy of the LUMO level being more than 2 eV below that of the Ar 4$s$ resonance in our X-ray absorption and core hole clock measurements, the spatial extent of the K-doped LUMO is considerably smaller than that of the photoexcited state shown in Fig. 3.

Moreover, and as described in more detail in Supp. Note 10, we find evidence for what we consider back-donation of electron density from the C$_{60}^-$ cage to the K$^+$ ion. Supp. Fig. 9a, b show two occupied molecular orbitals that have significant mixing between potassium and the cage, and that also incidentally have $A_g$ symmetry in the $\mathcal{I}_h$ point group. This effect is noticeably absent in the excited state of Ar@C$_{60}$, where apart from the distorted 4$s$ molecular orbital, all occupied molecular orbitals reside either entirely on the Ar atom or entirely on the C$_{60}$ cage.

### Separated, but connected: Ar@C$_{60}$/Ag(111)

Arguably the most compelling experimental evidence for mixing of the photoexcited Ar 4$s$ state with the surrounding fullerene cage comes from our measurements of a chemisorbed monolayer of Ar@C$_{60}$ on Ag(111), in concert with ground state periodic projector augmented wave DFT (PAW-DFT) calculations (see "Methods" and Supp. Note 11.) We first focus on the measurement of the Ar atom position with respect to the Ag(111) surface via the X-ray standing wave technique (Fig. 4). NIXSW is an exceptionally powerful probe of adsorbate geometry[20], and is especially well-suited for endofullerene systems. Two key parameters result from an NIXSW measurement: the coherent fraction, $f_c$, a measure of the level of order in the adsorbate positions, and the coherent position, $p_c$—the position of the adsorbate with respect to the substrate scattering plane.

Our deposition protocol (see "Methods") results in a value of $f_c$ for the encapsulated argon in the Ar@C$_{60}$ monolayer that is close to unity: 0.92 ± 0.05, signifying a highly ordered molecular layer. The value of 5.54 ± 0.04 Å for the argon atom height above the Ag(111) surface determined from the NIXSW analysis (Fig. 4) is identical to both the value predicted by our PAW-DFT calculations (5.55 Å; see Fig. 4, "Methods", and Suppl. Note 11) and the 5.5 ± 0.1 Å found by Pussi et al.[48] (from a LEED analysis) for the centre of the (empty) C$_{60}$ cage in the metastable 6:6-bond-down, on-top adsorption geometry. Our monolayer preparation method very much favours adsorption in this kinetically limited state. As such, the experimentally measured

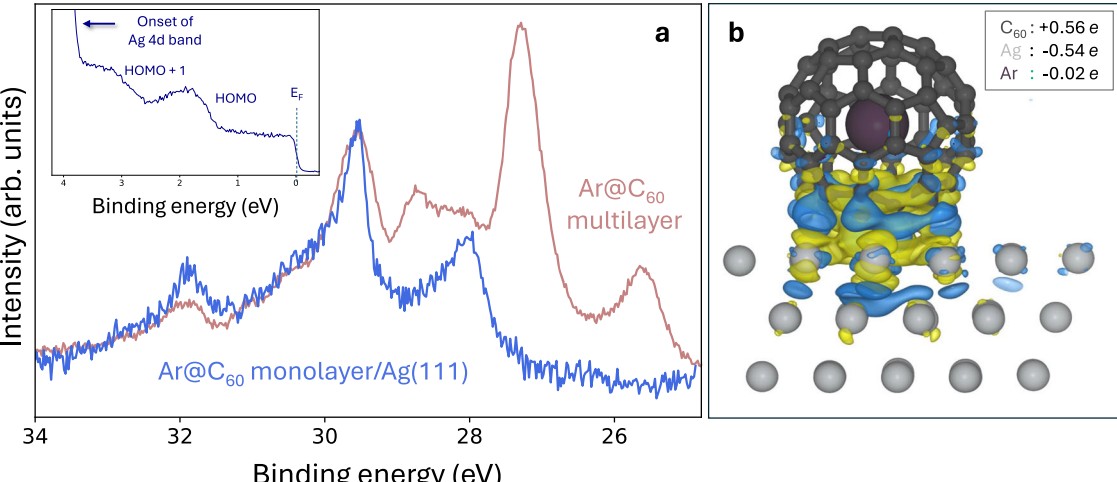

**Fig. 5 | Escape in less than a femtosecond: charge transfer for a chemisorbed Ar@C$_{60}$ monolayer. a** On-resonance deexcitation spectrum (in blue) for Ar@C$_{60}$/Ag(111) following a linear background subtraction and a shift to 1.25 eV higher binding energy so as to align with the corresponding spectrum for the multilayer sample (purple). There is a complete absence of spectator peaks for the Ar@C$_{60}$/Ag(111) sample, and thus a sub-femtosecond Ar 4$s$ delocalization time. We estimate an upper limit of 500 attoseconds (see text). **Inset:** Valence band spectrum ($\hbar\omega = 110$ eV) for the ∼1 ML Ar@C$_{60}$/Ag(111) sample. The HOMO and HOMO+1

binding energies exactly match those for 1 ML of empty C$_{60}$ on Ag(111)[65,66]; **b** Ground state DFT calculation showing the difference in charge distribution for Ar@C$_{60}$ adsorbed on Ag(111), as compared to the isolated molecule and metal. The vast majority of the charge difference is restricted to the fullerene cage-Ag(111) interface; the ground state electron density of the Ar atom is almost entirely unaffected by adsorption. (Blue: depleted charge; yellow: gained charge. Isosurface: $7 \times 10^{-4} e/$Å$^{-3}$.). Source data are provided as Source Data files.

5.54 ± 0.04 Å Ar-Ag(111) separation places the argon atom at the centre of the adsorbed endofullerene, its intracage position unperturbed by the chemisorption of the surrounding molecule.

For the metal-adsorbed Ar@C$_{60}$ monolayer, all trace of the spectator channel is removed and only the traditional Auger de-excitation pathway remains (Fig. 5a). The complete absence of spectator signal above the signal-to-noise ratio (SNR) limit of our experimental measurement means that the electron delocalization time is now at the sub-femtosecond level. Taking the magnitude of the measurement SNR into account[26], we can place an upper limit on the value of $\tau_D$. We first determine the standard deviation, $\sigma$, of the background noise in the binding energy region (25–27 eV) where we would expect spectator intensity to be located if it were present. Our criterion for signal detection is that the peak intensity should be a minimum of $3\sigma$ above the background. On this basis, we find that the minimium detectable spectator signal would be a factor of 0.08 smaller than the normal Auger-Meitner intensity. (We note that this is very close to the factor of 0.1 estimated by Föhlisch et al.[38] for the lowest practically resolvable charge transfer time available via the core hole clock method.) As such, we estimate that the upper limit of the electron delocalization time for the Ar@C$_{60}$ monolayer is $0.08\tau_{CH}$, i.e., ∼500 attoseconds. (Given the method of estimation, it is appropriate to quote only to 1 significant figure).

This value is more than an order of magnitude smaller than for an argon atom adsorbed directly on the Ag(111) surface[38] (where the equilibrium adsorption height, namely 3.3 Å[49], is over 2 ångstroms lower than that for argon in Ar@C$_{60}$). In other words, rather than acting to decouple the Ar 4$s$ excitation from the surrounding metallic environment (and thus impede the delocalization rate), we instead see the same effect, now accentuated, as for the bulk Ar@C$_{60}$ film: the fullerene cage provides a remarkably efficient conduit for electron transfer.

Ground state PAW-DFT calculations (Fig. 5b) predict substantial charge transfer between the Ag(111) surface and the fullerene cage (∼0.56$e$, to be compared with the 0.5$e$ determined in a previous DFT study[50] and 0.75$e$ estimated from photoemission measurements[51]). However, the charge state of the encapsulated argon remains essentially unaffected by chemisorption of the fullerene cage. (The SI

includes a discussion of the Ar 2$p$ and C 1$s$ photoemission spectra for the Ar@C$_{60}$ monolayer, which are consistent with this interpretation.) It is clear, therefore, that the very significant enhancement of electron delocalization rate for the endofullerene monolayer again arises from excited state coupling of a diffuse Ar-fullerene hybrid orbital with the Ag(111) electronic structure, rather than an adsorption-induced shift in the position of the argon atom or its electronic structure.

The calculations of Gauyacq and Borisov[37] can be used to determine the charge delocalization time of an hypothetical bare argon atom adsorbed at the same height of 5.5 Å as we measure for argon in Ar@C$_{60}$ on Ag(111). This is ∼65 fs, at least two orders of magnitude slower than the rate observed for the metal-adsorbed Ar@C$_{60}$ endofullerene. Moreover, SAMO-derived bands are robust against fullerene adsorption on metals[24,42]—indeed, SAMO states were first observed in a C$_{60}$ monolayer on Cu(111)[24]—and so the absence of any spectator contribution to the de-excitation spectra of the Ar@C$_{60}$ monolayer is consistent with a coupling of an Ar-SAMO hybrid orbital to the electronic reservoir of the Ag(111) substrate.

The surprising result that emerges from our study is that encapsulating an inert atom in a closed carbon cage yields a substantially enhanced level of electronic coupling to the environment. We measure electron delocalization times that are at least an order of magnitude faster for Ar@C$_{60}$ than for a bare argon atom, despite the absence of ground state mixing of the frontier orbitals of the fullerene with the encaged argon. Our results are consistent with electron transport via diffuse hybrid Ar-fullerene orbitals, in which the vast majority of the electron density is found outside the cage. This has intriguing implications with regard to controlling the chemistry of endohedrally-caged atoms via delocalized hybrid orbitals. Adding submolecular spatial resolution to the XAS measurements via a strategy similar to that introduced by Ajayi et al.[52] is of particular future interest in this regard.

## Methods
### Synthesis of Ar@C$_{60}$
Ar@C$_{60}$ was synthesised by molecular surgery[53], a process in which chemical reactions are used to open a hole in the C$_{60}$ cage large enough to allow argon to enter. A further series of reactions is then

used to close the hole to reform the pristine $C_{60}$ cage, which now contains an argon atom[15]. (Previously, $Ar@C_{60}$ has been obtained in very low yield by exposure of $C_{60}$ to argon at high temperatures and pressures followed by extensive purification (see, for example, Saunders et al.[14]).

## Preparation of multilayer and monolayer films of $Ar@C_{60}$

The Ag(111) surface was first cleaned via repeated sputter-anneal cycles (1 keV $Ar^+$ ions at an argon pressure of ~$2 \times 10^{-5}$ mbar; sample annealing temperature ~550 °C) until a sharp (1 × 1) low energy electron diffraction (LEED) pattern was visible and there was no evidence of C 1$s$ or O 1$s$ core-level signals in photoemission spectra (for which the photon energy was tuned to maximise the surface sensitivity of the photoelectrons.) $Ar@C_{60}$ was then deposited from a thermal evaporator operating at a temperature of 400 (±20) °C onto the Ag(111) sample, which was held at a temperature of ~180 K throughout the deposition in order to prohibit reconstruction at the fullerene-Ag(111) interface[54]. This produced a $(2\sqrt{3} \times 2\sqrt{3})R30°$ LEED pattern[54].

Formation of monolayer coverages in this way essentially "freezes out" reconstruction (via "nanopitting"[48,54]) of the Ag(111) substrate, resulting in an X-ray standing wave coherent fraction value close to unity, substantially larger than that previously observed for endofullerene monolayers on Ag(111)[55], due to the homogeneity of the molecular adsorption sites. (Note, however, that a small "overshoot" in molecular coverage beyond the first monolayer is difficult to avoid using this protocol—see Supp. Note 4) Multilayer coverages of sufficient thickness to quench photoemission signal from the Ag(111) substrate required cumulative deposition times of order four to five hours. A shift of ~0.4–0.5 eV in the C 1$s$ core-level binding energy for monolayer vs multilayer coverages (Supp. Note 4), coupled with a measurement of the ratio of the intensities of the C 1$s$ and Ag 3$d$ photoemission peaks, facilitated the identification of monolayer (and close-to-monolayer) coverage.

## Photoemission, X-ray absorption, and NIXSW measurements

All experimental work described in this paper was carried out at beamline I09, *Surface and Interface Analysis*, at the Diamond Light Source[56]. I09 is equipped with both a hard X-ray undulator, which was used for our NIXSW measurements, and a soft X-ray undulator, used for the acquisition of high-resolution C 1$s$, Ar 2$p$, Ag 3$d$, and valence band photoemission spectra, and for Ar $L_{2,3}$ and C K-edge X-ray absorption spectroscopy. (The resolving power of the soft X-ray branch is 10,000).

**Circumventing beam damage.** Considerable care was taken to reduce beam damage by detuning the beam (i.e., applying a small change in the undulator gap value to reduce peak intensity) and cooling the sample to temperatures between 100 K and 180 K. In previous synchrotron-based work—both published[55] and unpublished—on endofullerene samples, we have found that measurements acquired at room temperature and without any adjustment of the undulator output flux (and/or sample position) can result in significant beam damage. X-ray absorption and photoemission peaks would, at best, diminish in intensity on a timescale of minutes. We note that DiCamillo et al.[57] have reported similar beam damage observations, i.e., the loss of Ar 2$p$ signal, in their lab-based X-ray photoelectron spectroscopy studies of $Ar@C_{60}$. Conversely, Morscher et al.[16] instead did not observe depletion of argon from $Ar@C_{60}$ under either Mg K$\alpha$ or He I radiation.

Our approach to minimising beam damage for $Ar@C_{60}$ samples involved (a) acquiring spectra at low sample temperatures (a maximum of 180 K), and (b) detuning the undulator so as to reduce the photon flux on the sample by an order of magnitude. Throughout the beamtime experiments we regularly checked for evidence of beam damage by comparing photoemission and X-ray absorption peak intensities. No degradation of signal intensity, or other characteristics such as lineshape, was observed for either soft X-ray (photoemission, X-ray absorption (XAS), resonant Auger/photoemission) or hard X-ray (NIXSW) spectroscopies.

**NIXSW measurements and analysis.** NIXSW data (Fig. 4) were acquired via the accumulation of Ar 2$p$ photoemission spectra during twelve separate sweeps of photon energy through the Ag(111) Bragg condition. (At 180 K the bulk lattice constant for Ag is 4.0779 Å, equating to a (111) plane spacing of 2.354 Å and a corresponding Bragg energy of 2633.47 eV). A Jupyter Notebook version of NIXSW/dynamical X-ray scattering code that had previously been developed by two of the authors (DAD and T-LL) was used to fit the data and extract the coherent position and coherent fraction parameters.

## Core hole clock considerations and fitting

The determination of the delocalization/charge transfer time, $\tau_D$, is dependent on accurate knowledge of the core hole lifetime, $\tau_{CH}$. Our choice of $Ar@C_{60}$ (as opposed to other endofullerenes) for the measurement of intracage excited state delocalization was motivated in part by the ready availability of high precision measurements of the Ar 2$p$ core-hole lifetime (5.7 ± 0.1 fs[33]). Moreover, argon is a particularly attractive target species for CHC experiments due to the easily-resolved spectator shift, i.e., the difference in kinetic energy between the electron spectra arising from the two distinct decay channels in Fig. 1b. The delocalization time, $\tau_D$, is calculated from the relative integrated intensities of the spectator and "traditional" Auger-Meitner contributions ($I_{Spec}$ and $I_{Auger}$, respectively) to the de-excitation spectra:

$$\tau_D = \left( \frac{I_{Spec}}{I_{Auger}} \right) \tau_{CH} \qquad (1)$$

An additional motivation for the use of argon lies in the X-ray absorption linewidth, and, in particular, its relationship to the resolving power of the beamline (~10,000) at the Ar $L_3$ edge. As noted above, our measurements were acquired in the Auger-Meitner resonant Raman mode, for which the X-ray photon bandwidth (~25 meV in this case) is significantly smaller than the natural lifetime of the core-hole. When this is the case, the kinetic energy of the spectator peaks tracks the variation in photon energy across the absorption edge (for the reasons discussed by Menzel[26]); in other words the spectator peaks remain at fixed binding energy (see Fig. 2). Conversely, the peaks arising from the "traditional" (i.e., non-Raman) Auger-Meitner process remain at fixed kinetic energy. These, and many other, constraints were applied during the fitting of the set of Auger-Meitner decay spectra acquired across the X-ray absorption resonance. An extensive description of our fitting strategy is given in the supplementary information.

## Density functional theory (DFT) calculations

For the results shown in Fig. 3, the structure of $Ar@C_{60}$ was first optimised at the DFT/PBE/6-31++G** level of theory and the orbitals involved in the Ar 2$p$ → 4$s$ transition were identified. (Atomic coordinates are provided as Supplementary Data 2). The excited state was then calculated at the same level of theory, within the Q-Chem 5.4 package[58], with the aid of the MOM[22,23] (see following section) to maintain the core-hole during the DFT calculation. The ground state structural and charge transfer calculations of Fig. 3 were carried out with the Vienna Ab initio Simulation Package[59], under periodic boundary conditions and within the plane-wave projector augmented-wave (PAW) method[60]. The Ag(111)-$(2\sqrt{3} \times 2\sqrt{3})R30°$-$C_{60}$ structures were optimized using the local spin density approximation (LSDA) with a force tolerance of 0.01 eVÅ$^{-1}$ and an electronic convergence criterion of $10^{-6}$ eV. The energy cut-off was set to 500 eV, and a

Monkhorst-Pack $k$-point grid of $3 \times 3 \times 1$ was used to sample the Brillouin zone. Electronic charges associated with individual atoms, utilized in the calculations of charge transfer, were derived using Bader analysis[61]. The atomic visualisations were generated using the Open Visualisation Tool[62]. As described in Supp. Note 11, a variety of other DFT methods and adsorption geometries were employed to determine the Ar-Ag(111) separation but each provided poorer agreement with the NIXSW measurements than the LSDA approach.

### The maximum overlap method (MOM)

The MOM provides an efficient approach for calculating excited states by modifying the orbital selection step in the SCF procedure and targeting solutions with non-Aufbau occupations from a ground state reference set of molecular orbitals[22,63]. By employing a simple orbital overlap-based criterion, the MOM prevents the variational collapse to the lowest energy solution. The MOM begins with an initial set of molecular orbitals (MOs) generated from the ground-state configuration of the system. Excitations are then introduced by modifying the occupation patterns, typically replacing one or more occupied orbitals with virtual orbitals. At each SCF iteration, the MOM algorithm applies an overlap metric to select the occupied orbitals that are most similar to the target orbitals from the previous iteration, guiding the SCF solver towards the intended excited state. In this work, we first calculated the electronic structure of Ar@$C_{60}$ at the PBE/6-31++G** level of theory to identify the relevant Ar-based $2p$ and $4s$ molecular orbitals. We then constructed an initial guess for the target excited state of this system by promoting an electron from one of the occupied Ar-based $2p$ molecular orbitals to the unoccupied Ar-based $4s$ molecular orbital. Subsequently, we used the MOM to relax the occupied molecular orbitals while staying as close as possible to the initial pattern.

## Data availability

Source data are provided with this paper. All raw data generated in this study have been deposited in the University of Nottingham Research Data Management repository at https://doi.org/10.17639/nott.7457 Source data are provided with this paper.

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

## Acknowledgements

We acknowledge very helpful discussions with Andrey Borisov and thank Dave McCue, Senior Beamline Technician, Diamond Light Source for his technical assistance and expertise. The synthesis of Ar@$C_{60}$ was supported by EPSRC (UK) (EP/M001962/1 and EP/P009980/1), R.J.W., M.H.L.), and the European Research Council (786707 - FunMagResBeacons, R.J.W., M.H.L.). We gratefully acknowledge Diamond Light Source for the provision of beamtime via allocations SI-31574-1, SI31574-2, and SI-31574-3 (PI:PM), and for the award of PhD studentship STU0425 (C.F.). P.M. thanks the Engineering and Physical Sciences Research Council (EPSRC) for an Established Career Fellowship (EP/T033568/1). E.B. acknowledges a Royal Society Wolfson Fellowship. The computational work has been supported by the EPSRC Programme Grant 'Metal Atoms on Surfaces and Interfaces (MASI) for Sustainable Future' (EP/V000055/1, E.B.) and the University of Nottingham's Augusta HPC service (E.B., B.H.) and the Sulis Tier 2 HPC platform funded by EPSRC Grant EP/T022108/1 and the HPC Midlands+ consortium (B.H., E.B.). D.A.D. acknowledges a New Investigator Award from the Engineering and Physical Sciences Research Council [EP/X012883/1].

## Author contributions

C.F., F.J., T.J., S.A.P., D.A.D., Y.W., B.K., J.N.O.S. and P.M. carried out the experiments during three beamtime allocations at I09, Diamond Light Source; A.F., S.G., I.P., B.H. and E.B. were responsible for the D.F.T. and M.O.M. calculations; S.B., G.H., M.W., M.L.H. and R.J.W. provided the endofullerene samples; T.-L.L. provided key advice and expertise related to the beamline experiments; C.F. and P.M. were responsible for experimental data analysis; P.M. drafted the paper, with input and feedback from all co-authors.

## Competing interests

The authors declare no competing interests.
