## [Transparent Peer Review File · Nature Communications]

Timing the Escape of a Caged Electron

Corresponding Author: Professor Philip Moriarty

Version 0:

Reviewer comments:

Reviewer #1

(Remarks to the Author)

The manuscript by Fields et al details core hole clock measurements performed at the Diamond Light Source on Ar endohedral fullerenes (Ar@C60). In the experiments, an electron in the Ar atom is excited to a 4s orbital and the resulting decay and emission of photoelectrons is measured as a function of kinetic energy and X-ray energy. Careful analysis of the spectra in the energy domain allows for determination of the fraction of 4s electrons that escape the initial core-hole excited state and can give a measure of how fast the electron is transported away, in this case from the Ar ion. The authors find surprisingly that the electron transfer from Ar@C60 on a Ag(111) surface is faster than even an isolated Ar atom on the same surface. This is really interesting and has implications for molecular electronics and a wealth of other subjects where electron transfer is important (such as biology and photovoltaics etc).

The paper is very well written, and all assumptions and conclusions drawn from the data are clearly presented and discussed. The fast electron transfer is attributed to a mixing of the 4s Ar orbital with superatom molecular orbitals (SAMOs) on C60. These are very interesting and have been studied quite a lot recently. It is known that these molecular orbitals in monolayers of C60 form highly delocalised states that extend throughout the solid and increase conductivity. The authors use a range of techniques (incl. computational) to demonstrate the involvement of the SAMOs. The experimental parameters they obtain are all carefully compared to experiments and calculations in the literature on comparable systems such as C60 or Ar on graphite/Ag surfaces. All the results agree with the literature, but I want to stress that the novelty of this work is that the SAMOs produce a very fast delocalisation and subsequent transport away of the excited electron to the metal substrate. This is even faster than Ar on its own, so the mixing with the SAMOs is really speeding up the processes.

The methodology is sound and all the details provided enable reproduction of the data. Because the paper is so well written and the conclusions so well argued, I recommend publish as is. The results really are spectacular and I fully believe that Nature Comm is the right journal for this work. Congratulations to the authors!

Reviewer #2

(Remarks to the Author)

The authors in their paper entitled Timing the Escape of a Caged Electron investigate the timescales of electron delocalization in the Ar@C60 system. They employ the core-hole-clock method instead of time-resolved femtosecond spectroscopy which is a smart way to read the information about the delocalization time from energy-resolved spectra. The results show that the delocalization time in Ar@C60 film is 6.5 fs and if the same system is deposited on 2D monolayer of Ag, the delocalization is even faster ~ 600 at. These results are noteworthy and of significance because charge-transfer processes between molecules on metal surfaces are important in design of variety of electronic devices. In this particular case, the electron which is transferred comes from the argon atom inside the fullerene cage, which is novel. Several related studies has been reported in the last few years (for example <https://doi.org/10.1021/acs.accounts.0c00627>) based on the same concepts, so the use of core-hole-clock method is not entirely new. The presented data support the conclusions, but additional details should be provided, mainly in the theoretical section which only vaguely supports the experiment (even though I understand that the size of the studied complex prevents advanced ab initio methods). The analysis does not contain obvious flaws, interpretation and conclusions drawn are sound. The work meets in general expected standards; however, I believe that the theoretical section would greatly benefit from additional calculations and that the methodology could be extended, see below.

In general, I support publication since the topic is timely and significant for the broad readership. However, I see several rather major issues which should be addressed with care.

1. The Introduction section is clearly written and contains all details. The Results section starts with Figure 1 which is a bit too complex. Label (b) is placed in the lower right corner, all other letters in the upper corner, please unify. The panel ii) Auger decay in fact represents the charge-delocalization part. The delocalized electron is hard to spot. In panel (e) it is not easy to identify the linearly dispersive peaks, could you please show them with for example white dashed lines and connect to the peaks in panel (f)? Panel (d) contains the valence spectrum and Ar 2p_{3/2} line but is it necessary in the Figure?

2. In the Introduction and Results section the term tunnelling is used for electron delocalization. Which kind of tunneling you refer to?

3. On p. 5 when discussing the delocalization times, the authors claim that there is no mixing of frontier argon and fullerene orbitals and that the charge transfer is presumable slow. Could you provide some quantitative information, i.e. on orbital mixing, is it 0 or a few percent?

4. In Figure 2, two 4s-SAMO orbitals are shown, the isosurface values are in the Figure 2 caption. I believe that one picture is ok to convey the information. Could the authors also break down which orbitals participate in the diffuse 4s-SAMO the most? The RDF picture is nice and instructive!

6. Theoretical interpretation of the results on Ar@C60 is based on the MOM calculations. The studied excitation is 2p_{3/2} → 4s and the MOM method cannot account for the splitting of the 2p orbitals into 2p_{3/2} and 2p_{1/2}, since no relativistic effects are included. In the text, it is not clear, how you constructed the excitation. Could you please clarify if excitation from both 2p_{3/2} orbitals were calculated? The energy of these transitions via MOM are also heavily affected by not including the relativistic effects, the respective energies are nor provided nor commented. Could the authors explain why they opted for the MOM method which has so many drawbacks? I think that TDDFT calculations may be a more suitable option since they address relativistic effects and allow for an advanced wave function analysis (for example using the THEODORE code). Could the author provide comparison of the TDDFT calculations and the MOM results? In the MOM calculations the final wavefunction is relaxed, on the other hand, TDDFT excitations are unrelaxed, does it change the overall picture?

7. In the case Ar@C60 is located on the Ag surface, the predicted delocalization time is <~600 as stated in the Introduction section. However, the Results section show no data supporting the claim, how was the delocalization timescale derived? This section lacks details for the work to be reproduced.

I've spotted two typos:

p. 9, Figure caption, "C60 on Ag(111)[? ?]" reference is clearly missing.

p. 5, tau_D, D should be subscript

Reviewer #3

(Remarks to the Author)

The manuscript "Timing the escape of a caged electron" by Fields et al presents a mainly experimental study of the delocalization timescale of a core-excited state of Ar in C60. The experiments appear to be well done, and the results are interesting. For the manuscript to deserve publication in Nature Communications, however, the authors need to satisfactorily address a number of points, see below.

Ar in C60 is an interesting but odd system. In what way are the presented results relevant for other systems?

On page 2, the authors state that the investigated system is unique. Are there any similarities to other cage-like structures, such as chlorates? Since Ar in C60 is a rather odd system, to what extent are the results of general interest if the system is also unique?

The authors also state that there is no hybridization between argon and the surrounding C60 in the ground state. This is not strictly correct, and a few lines further down, the authors even discuss a hybrid orbital. The ground state hybridization may be small, and may not cause the observed effects, but there is always hybridization. This needs to be clarified.

It is also stated that "one might expect that the lack of overlap would yield long-lived electronically excited states inside the cage". I do not agree that this is to be expected. Core-excited argon is very different from ground-state argon. According to the equivalent core, or Z+1, approximation, argon in the 2p⁻¹ 4s state is similar to ground-state potassium. Calculations of the electronic structure of potassium in C60 show that the 4s orbital is delocalized on the C60, with the potassium essentially being a K⁺ ion inside the C60 cage, see e.g. Östling and Rosén, Chemical Physics Letters 202, 389 (1993). This makes the rapid delocalization of the excited electron less surprising, and a discussion of the relation between Ar 2p⁻¹ 4s and K could be an interesting addition to the manuscript.

The main result of the manuscript is based on a measurement including both resonant Auger and electron delocalization. The complete process is thus quite complex, and for most readers it would be helpful with a schematic explanatory figure.

As I understand, the results are based on the spectator transitions. Where participator transitions observed? What could be learned from them, or their absence?

The experimental results are interpreted in terms of the core-hole clock. In this context, would be good with a discussion of the underlying assumptions, and also references to original papers, not only later reviews.

What could simulations tell about the delocalization time scale?

Figure 1 presents the experimental results in subfigure 1e. It also contains many other subfigures, which makes it very crowded.

Subfigure 1b shows schematics of the excitation and the local decay processes. How are those related with the delocalization process?

Subfigure 1d shows the Ar 2p XPS spectrum as an inset. The Ar 2p binding energy is lower than the photon energy used in the resonant Auger measurement. Does this mean that the Ar $2p^{-1}$ state is lower in energy than the Ar $2p^{-1} 4s$ state? What consequences does that have for the discussion?

Subfigure 1e shows the Auger map, which is the main result. This should be enlarged, and guiding lines helping the reader to identify the different processes should be added.

Subfigure 1f shows an example of an Auger spectrum, decomposed into contributions. It would be interesting to see spectra for at least two different photon energies exhibiting different "Auger/spectator" ratios

Subfigures 1e and 1f have similar colour scales, but the colours don't mean the same thing in the two.

Figure 2 shows the "4s-SAMO" hybrid orbital.

What are the relative contributions to the "4s-SAMO" hybrid orbitals? How much is C60, how much is Ar?

Why is the "4s-SAMO" orbital so p-like?

For comparison, it would be interesting to plot the ground state in the same way.

The core-hole clock provides indirect temporal information. What are the prospects of directly measuring the delocalization dynamics?

Reviewer #4

(Remarks to the Author)

Version 1:

Reviewer comments:

Reviewer #2

(Remarks to the Author)

The authors have taken all the comments seriously and implemented them in the revised text and I am happy that the whole review process helped to improve the manuscript so much.

Specifically, Figure 1 is now clear, the authors made the effort to clarify important parts. The delocalized electron is now super easy to spot. Figure 2 is updated and the figure is perfect, easy to interpret and straightforward. I am also happy that revising the figures helped the interpretation - now the delocalization times with uncertainty related issues are thoroughly discussed which makes the paper sound. I acknowledge the explanation of tunneling and more importantly, I appreciate the population analysis (and I understand that the authors must have spent considerable amount of time on it) which brings quantification of the delocalization. I also agree with improvement of the SI with respect to TDDFT calculations. Overall, all my comments were addressed and all my concerns were answered and I fully support the publication of the paper.

Reviewer #3

(Remarks to the Author)

The authors have addressed the reviewer comments satisfactorily. They have made substantial revisions to the manuscript and supplemental information, particularly addressing key concerns about hybridization, the $Z+1$ approximation, and the clarity of the figures. This has improved the manuscript, making it more accessible to readers and providing clearer explanations of the theoretical and experimental parts of the work. I would therefore recommend accepting the manuscript for publication.

Reviewer #4

(Remarks to the Author)

RESPONSE TO REVIEWERS

We are very grateful to all four reviewers for the time and effort they took in reading and reviewing our paper, and for returning in-depth, insightful, and valuable feedback and criticisms. Our paper has been strengthened considerably by addressing the reviewers' concerns. Below, we respond to each of the reviewers' comments in turn, highlighting where the manuscript and supplementary information file have been revised to address the reviewers' concerns. Note that all additions/revisions are highlighted in red in the revised manuscript and SI (and in the responses below). We also note that we have replaced all instances of "Auger" in the original manuscript with "Auger-Meitner", for the reasons discussed in D. Matsakis *et al.*, *Physics Today* **72**, 10 (2019).

Reviewer #1

We were delighted to read that Reviewer #1 believed the paper should be published as is, was "very well written", and that the results "really are spectacular". Our sincere thanks to the reviewer for reading and assessing our paper, and for their positive appraisal of our work.

Reviewer #2

Reviewer #2 supports publication, stating that the results "are noteworthy...novel...timely and significant", and that the conclusions we have drawn "are sound", but raises a number of important concerns, which we have aimed to address as comprehensively as possible in the revised manuscript and SI.

"Several related studies have been reported in the last few years (for example, <https://doi.org/10.1021/acs.accounts.0c00627>)..."

This is a review article of direct relevance to our work, which we now cite in the revised manuscript. We thank the reviewer for bringing it to our attention.

1. "The Results section starts with Figure 1, which is a bit too complex..."

We agree. In hindsight, the original Fig.1 was much too "busy". We have therefore **split the original Figure 1 in two** so that the schematic illustrations of the X-ray excitation and subsequent Auger-Meitner decay processes now form Fig. 1 by themselves, with the experimental spectra now comprising Fig. 2. This reduces the complexity of the opening figure in the paper and improves the readability of the manuscript.

"...the delocalized electron is hard to spot."

Agreed. **We have modified the illustration in Fig. 1** so that the delocalised electron is much more visible.

“It is not easy to spot the linearly dispersive peaks...”

Once again, we agree with the reviewer.

We have now not only highlighted/delineated the normal Auger-Meitner and spectator peaks using dashed lines in Figure 2, as the reviewer suggested, but have (a) revised the color map (and overlaid a contour map) to accentuate the features in the resonant Auger-Meitner map so as to make them easier for the reader to “track”, and (b) added a lengthy section to the SI that describes in detail our fitting strategy.

Note that in revising the paper, we found that the uncertainties in delocalisation times, τ_D , that we quoted in the originally submitted paper were significantly under-estimated; non-linear least squares fitting algorithms combined with a heavily constrained broad parameter space are prone to returning significant underestimates for the uncertainties in the fits via the traditional approach (i.e. diagonalisation of the covariance matrix) [see, for example, Andrae, Schulze-Hartung, and Melchior, “Dos and don’ts of reduced chi-squared”, arXiv:1012.3754 (2010); <https://arxiv.org/pdf/1012.3754>]

While these uncertainty-related issues are common to non-linear least squares fitting in general, we also discovered a specific issue with our use of the Python LMFIT package for fitting: LMFIT’s interpretation of the term “amplitude” in peak-fitting is far from standard (see, as just one example, the extended discussion at <https://github.com/lmfit/lmfit-py/issues/155>). Constraints on peak amplitudes must therefore be handled with great care otherwise LMFIT will not hold relative intensities constant without also adjusting peak widths.

To circumvent this issue, for the resubmitted version of our manuscript we have revised our fitting code so as to constrain peak (relative) heights rather than peak amplitudes. This provides much more robust fits. Although this change did not affect the on-resonance value of the delocalization time extracted from the fits, nor the trend in delocalization time with photon energy, there was some “redistribution” of spectral intensity between the normal Auger-Meitner and spectator components.

We have revised the experimental uncertainties and fitting strategy accordingly and **discuss this at length in Section 4 of the SI.**

“Panel (d) contains the valence spectrum and Ar 2p_{3/2} line, but is it necessary in the figure?”

The valence band and core-level spectra were indeed rather misplaced in the original Fig. 1. We have now **moved them to Fig. S1 and Fig. S2 of the SI.**

“2. In the Introduction and Results sections the term tunnelling is used for electron delocalisation. What kind of tunnelling are you referring to?”

We use the term tunnelling here in the context that it is applied throughout the core hole clock literature (see, for example, the reviews by Bruhwiler et al, Menzel, and Zharnikov, references 19, 26, and 28, respectively, in the main paper). The interpretation is generally in the context of WKB-type tunnelling through a barrier arising from the adsorbate-substrate interaction. There is no requirement for a resonant tunnelling process in the standard core hole clock approach, if this is what the referee has in mind. Typically, a broadened resonance arising from the adsorbate level of interest (in this case, the Ar 4s state) overlaps with a quasi-continuum of states due to the band structure of the substrate (in line with the Anderson-Newns model of adsorbate-substrate interaction.)

However, and as discussed at length by Föhlisch *et al.* [Ref 39 in the main paper], a simple “WKB-like” tunnelling model would predict an exponential decrease in the delocalisation time, i.e. an exponential increase in tunnelling rate, as the photon energy is increased (and thus the effective barrier height experienced by the photoexcited electron decreases.) In many cases (but not for Ar@C₆₀), precisely the opposite dependence is observed. Föhlisch *et al.* have attributed this to the role of wavevector matching.

“3. On p5 when discussing the delocalisation times, the authors claim that there is no mixing of frontier argon and fullerene orbitals and that the charge transfer is presumably slow. Could you provide some quantitative information, i.e. on orbital mixing, is it 0 or few percent?”

and

4. In Figure 2, two 4s-SAMO orbitals are shown, the isosurface values are in the Figure 2 caption. I believe that one picture is OK to convey the information. Could the authors also break down which orbitals participate in the diffuse 4s-SAMO the most? The RDF picture is nice and instructive!”

(In the following our response covers both points 3 and 4.)

We have spent a considerable amount of time addressing the reviewer’s questions re. mixing/orbital decomposition via a detailed set of calculations focussed on population and symmetry analysis. We are

grateful to the reviewer for raising the matter – addressing this point from a theoretical/quantum chemistry perspective has led to important new insights. To address the reviewer's comments as comprehensively as possible, we have: (i) revised Fig. 3, (ii) added a discussion of population and symmetry analysis to the manuscript (bottom of p.6 and p.7), and (iii) added two sections to the SI (#7 and #8) that provide more detail on our approach to symmetry/population analysis.

In light of these new calculations, “no mixing” for the ground state unoccupied orbital was too strong a statement in the original paper and we have changed this to “marginal” in the revised manuscript. As discussed in both the main paper (p7) and the SI, the ground state unoccupied 4s orbital is of 92% argon character whereas the excited state is predominantly (87%) of carbon character. This predominant carbon character is supported by the symmetry analysis also discussed on p7 of the main paper and the SI.

“6. Theoretical interpretation of the results on Ar@C₆₀ is based on the MOM calculations. The studied excitation is 2p_{3/2} -> 4s and the MOM method cannot account for the splitting of the 2p orbitals into 2p_{3/2} and 2p_{1/2}, since no relativistic effects are included. In the text, it is not clear, how you constructed the excitation. Could you please clarify if excitation from both 2p_{3/2} orbitals were calculated? The energy of these transitions via MOM are also heavily affected by not including the relativistic effects, the respective energies are nor provided nor commented.*

Could the authors explain why they opted for the MOM method which has so many drawbacks? I think that TDDFT calculations may be a more suitable option since they address relativistic effects and allow for an advanced wave function analysis (for example using the THEODORE code). Could the author provide comparison of the TDDFT calculations and the MOM results? In the MOM calculations the final wavefunction is relaxed, on the other hand, TDDFT excitations are unrelaxed, does it change the overall picture?”

Each of the perceptive points raised here by the reviewer deserved careful and extensive consideration. As such, we have carried out a large number of additional calculations and made significant revisions to the main paper and SI to justify our use of the MOM approach. A new section, *The maximum overlap method (MOM)*, has been added to *Methods* in order to describe our approach to implementing the MOM for Ar@C₆₀. Moreover, two new sections (#9 and #10) have been added to the SI that (i) compare the MOM and time-dependent DFT approaches (Table 2 in the SI is particularly instructive in this regard), and (ii) discuss the key difficulties with implementing a relativistic treatment.

♦ Note that the reviewer's report did not include a point 5.

“7. In the case Ar@C₆₀ is located on the Ag surface, the predicted delocalization time is <~600 at as stated in the Introduction section. However, the Results section show no data supporting the claim, how was the delocalization timescale derived? This section lacks details for the work to be reproduced.”

It was indeed remiss of us not to provide more detail on the estimation of the delocalisation time for the Ar@C₆₀/Ag(111) monolayer sample. We have now added a discussion (bottom of p.10/top of p.11 of main paper) that outlines how we use a 3 σ criterion to set a limit for detectable spectator intensity. Our approach is very much in line with previous estimates of the upper/lower timing limits of the core hole clock approach (see, for example, Ref. 26 of the main paper.) With this slightly more rigorous approach we have revised the estimate of the upper limit for delocalization from 600 attoseconds to 500 attoseconds. (As noted in the main paper, quoting the delocalisation time in this case to more than one significant figure is not warranted.)

“I've spotted two typos:

p. 9, Figure caption, "C60 on Ag(111)[? ?]" reference is clearly missing.

p. 5, tau_D, D should be subscript”

We thank the reviewer for spotting these typos, which we have now corrected.

Reviewer #3

We are similarly grateful to Reviewer #3 for raising a number of important concerns with our original manuscript. In addressing the reviewer's critiques, the paper has been strengthened considerably.

“Ar in C₆₀ is an interesting but odd system. In what way are the presented results relevant for other systems?”

On page 2, the authors state that the investigated system is unique. Are there any similarities to other cage-like structures, such as clathrates? Since Ar in C₆₀ is a rather odd system, to what extent are the results of general interest if the system is also unique?”

Ar in C₆₀ is indeed an “odd” system, and its oddness/novelty was a key motivation for our core hole clock experiments. As compared to other systems in which charge transfer has been studied via the CHC technique or ultrafast pump-probe spectroscopy, Ar@C₆₀ represents an important limiting case: the encaged argon, in the ground state at least, interacts exceptionally weakly with the surrounding fullerene cage. Yet, despite the minimal chemical interaction of the encapsulate with the surrounding chemical cage, the charge transfer rate determined via the CHC method is much faster than might be expected: for the monolayer Ar@C₆₀/Ag(111) system, the

delocalization rate is more than an order of magnitude faster than for a “bare” adsorbed argon atom.

As noted by Reviewer 1, this “*is really interesting and has implications for molecular electronics and a wealth of other subjects where electron transfer is important (such as biology and photovoltaics, etc.)*”. Similarly, Reviewer 2 highlights that our results are “*noteworthy and of significance because charge-transfer processes between molecules...are important in the design of a variety of electronic devices.*” The aspects of our work that are of general interest are not so much related to the unique structure of the endofullerene molecule itself; rather, it is the insights gained into the role of coupling/decoupling and orbital mixing/hybridisation (in both the ground and excited state) in electron transfer.

In an early draft of the manuscript we included a paragraph that outlined a comparison of endofullerenes with a variety of other cage-like structures, but removed this for reasons of space/word count before the original submission. **To address the reviewer’s comment, we have restored a sentence to the revised manuscript (p. 2) that highlights the key difference with other cage-like systems.**

“The authors also state that there is no hybridization between argon and the surrounding C60 in the ground state. This is not strictly correct, and a few lines further down, the authors even discuss a hybrid orbital. The ground state hybridization may be small, and may not cause the observed effects, but there is always hybridization. This needs to be clarified.”

The reviewer is correct to point out that we overstated the lack of hybridisation in the original manuscript, and we are grateful for their insight. They raise very similar concerns to those of Reviewer 2 on this particular point. As discussed in our response to points #3 and #4 of Reviewer 2, we have carried out a set of additional calculations and made extensive changes to the paper to address the reviewers’ criticisms re. the extent of hybridisation. To reiterate, we have (i) revised Fig. 3, (ii) added a discussion of population and symmetry analysis to the manuscript (bottom of p.6 and p.7), and (iii) added two sections to the SI (#7 and #8) that provide more detail on our approach to symmetry/population analysis.

*It is also stated that “one might expect that the lack of overlap would yield long-lived electronically excited states inside the cage”. I do not agree that this is to be expected. Core-excited argon is very different from ground-state argon. According to the equivalent core, or Z+1, approximation, argon in the $2p^{n-1} 4s$ state is similar to ground-state potassium. Calculations of the electronic structure of potassium in C60 show that the 4s orbital is delocalized on the C60, with the potassium essentially being a K^+ ion inside the C60 cage, see e.g. Östling and Rosén, *Chemical Physics Letters* 202, 389 (1993). This makes the rapid delocalization of the excited electron*

less surprising, and a discussion of the relation between Ar $2p^{-1} 4s$ and K could be an interesting addition to the manuscript.

This is a fascinating and critical point, and we have therefore spent a significant amount of time assessing the role of the equivalent core/Z+1 approximation in our CHC measurements, including carrying out a new set of DFT calculations focussed on this issue. A new section, *Role of the Z+1 approximation*, has been added to the manuscript (p.8) that describes our results in relation to K-doped C₆₀. Similarly, a section has been added to the SI (Section 11) that provides more information on the relationship of the Z+1 approximation and K-doped C₆₀ to our CHC results. We find that, while there are indeed parallels between K@C₆₀ and core-excited Ar@C₆₀, as highlighted by the reviewer, there are also significant differences in terms of both the spatial extent of the excited state orbital and mixing between the argon and fullerene cage electron density.

The main result of the manuscript is based on a measurement including both resonant Auger and electron delocalization. The complete process is thus quite complex, and for most readers it would be helpful with a schematic explanatory figure.

An explanatory figure along the lines suggested by the reviewer was included in the original manuscript as Fig. 1(b), but, as noted by both Reviewer 2 and Reviewer 3, Fig. 1 was overly complicated and obscured key details. As outlined in our response to Reviewer 2, we have split the original Figure 1 into two so that the schematic diagrams explaining the original X-ray excitation step and subsequent Auger-Meitner decay processes are now shown without any experimental data in Fig. 1. In addition, we revised the schematics so that the delocalized electron is clearer.

As I understand, the results are based on the spectator transitions. Were participator transitions observed? What could be learned from them, or their absence?

In the SI for the original submission, we had included a figure showing the negligible contribution of participator contributions. However, the reviewer's important question has prompted us to instead include a revised version of that figure in the main paper. In the revised manuscript, Fig. 2(c) shows a set of electron spectra around the resonance condition that include both the Auger-Meitner decay spectra and the Ar 3s photoemission peak. It is clear from those spectra that the Ar 3s spectral intensity does not resonate. This lack of participator contribution is important not just from the perspective of accurate estimates of the delocalisation time but is also interesting in the context of the reviewer's comments re. the Z+1 approximation and parallels with K-doped C₆₀.

The experimental results are interpreted in terms of the core-hole clock. In this context, would be good with a discussion of the underlying assumptions, and also references to original papers, not only later reviews.

Agreed. We have now added references to the original CHC papers (Refs. 18 and 29 of the revised manuscript), and noted the key assumptions (p.4 of revised version of main paper).

What could simulations tell us about the delocalisation time scale?

This is a very good question, which we discussed at length with Andrey Borisov, an expert in wave packet propagation (WPP) methods, in the early stages of analysing the electron spectroscopy/core hole clock data for Ar@C₆₀. (Borisov's important input is noted in the Acknowledgements section of the paper). A key requirement for WPP methods is a high-level quantitative description of the excited state wavefunction, which plays the role of the initial condition for the Cauchy problem. This remains a largely unsolved, non-trivial task for DFT methods [see Ref. 30 of the SI for a discussion], which becomes even more complicated for larger chemical systems such as Ar@C₆₀ and Ar@C₆₀/Ag(111). Although MOM-DFT might, in principle, be a good source of the initial wavefunction required for the WPP calculations, implementing a theoretical framework to solve this long-standing problem goes far beyond the scope of the present paper.

We have added a paragraph to Section 9 of the Supplementary Information to address this important point.

Figure 1 presents the experimental results in subfigure 1e. It also contains many other subfigures, which makes it very crowded.

As noted in our response to Reviewer 2, we have split off the schematic illustrations from the experimental data – both together comprised Fig. 1 in the original manuscript. The figures are now much less crowded.

Subfigure 1b shows schematics of the excitation and the local decay processes. How are those related with the delocalization process?

We have improved visibility of the delocalised electron in Fig. 1(b). See also our response to Reviewer 2 on this point.

Subfigure 1d shows the Ar 2p XPS spectrum as an inset. The Ar 2p binding energy is lower than the photon energy used in the resonant Auger measurement. Does this mean that the Ar 2p⁻¹ state is lower in energy than the Ar 2p⁻¹ 4s state? What consequences does that have for the discussion?

This point relates to the difference between the energy of the ionised and the excitonic states. Section 2 of the SI submitted with the original paper discussed this important issue. In the revised SI, this is now

Section 3 (but otherwise remains unchanged from the original submission.)

Subfigure 1e shows the Auger map, which is the main result. This should be enlarged, and guiding lines helping the reader to identify the different processes should be added.

Agreed. Reviewer 2 also made a similar comment. We repeat our response to Reviewer 2 here: We have now not only highlighted/delineated the normal Auger-Meitner and spectator peaks using dashed lines in Figure 2, as the reviewer suggested, but have (a) revised the color map (and overlaid a contour map) to accentuate the features in the resonant Auger-Meitner map so as to make them easier for the reader to “track”, and (b) added a lengthy section to the SI that describes in detail our fitting strategy.

Subfigure 1f shows an example of an Auger spectrum, decomposed into contributions. It would be interesting to see spectra for at least two different photon energies exhibiting different “Auger/spectator” ratios.

Fig. S3 shows five Auger-Meitner decay spectra taken with photon energies of 245.22 eV, 245.26 eV, 245.30 eV (on-resonance), 245.34 eV, and 245.38 eV, with accompanying fits to show the Auger/spectator ratios. We have also added a section to the SI (Section 4) that describes our fitting strategy in detail.

Subfigures 1e and 1f have similar colour scales, but the colours don't mean the same thing in the two.

This is a potentially confusing point that we must admit completely escaped us for the original submission! As noted above, we have completely revised the colormap and presentation of the data in the resonant Auger-Meitner map (Fig. 2(b) of the revised paper). We have also clarified in the figure caption the relationship of the map to Fig. 2(a) (the X-ray absorption spectrum) and Fig. 2(d) (the on-resonance decay spectrum).

Figure 2 shows the “4s-SAMO” hybrid orbital. What are the relative contributions to the “4s-SAMO” hybrid orbitals How much is C60, how much is Ar? Why is the “4s-SAMO” orbital so p-like? For comparison, it would be interesting to plot the ground state in the same way.

These are important issues -- Reviewer 2 had similar questions/concerns. For clarity, we reproduce here our response to points #3 and #4 of Reviewer 2, which also addresses Reviewer 3's

questions. We carried out a large number of additional calculations related to a population and symmetry analysis of the excited state. In the resubmitted version of the paper, we have: (i) revised Fig. 3, (ii) added a discussion of population and symmetry analysis to the manuscript (bottom of p.6 and p.7), and (iii) added two sections to the SI (#7 and #8) that provide more detail on our approach to symmetry/population analysis.

The core-hole clock provides indirect temporal information. What are the prospects of directly measuring the delocalization dynamics?

This question relates to the possibility of pump-probe measurements of Ar@C₆₀ excited states. This is certainly something of keen future interest to our groups, via. for example, free electron laser-based measurements and time-resolved two-photon photoemission measurements, but will involve a substantial additional experimental effort, and will, in particular, be subject to successful beamtime proposals and the vagaries of allocating time at large scale facilities. As such, pump-probe measurements of this type, although of future interest, are outside the scope of the paper.